# OGWO-CH: Hybrid Opposition-Based Learning with Gray Wolf Optimization Based Clustering Technique in Wireless Sensor Networks

**Rajakumar Ramalingam** [1], **Dinesh Karunanidy** [1], **Aravind Balakrishnan** [1], **Mamoon Rashid** [2,*], **Ankur Dumka** [3,4], **Ashraf Afifi** [5] **and Sultan S. Alshamrani** [6]

1 Department of Computer Science & Technology, Madanapalle Institute of Technology & Science, Madanapalle 517325, India
2 Department of Computer Engineering, Faculty of Science and Technology, Vishwakarma University, Pune 411048, India
3 Department of Computer Science & Engineering, Women Institute of Technology, Dehradun 248002, India
4 Department of Computer Science & Engineering, Graphic Era Deemed to be University, Dehradun 248007, India
5 Department of Computer Engineering, College of Computers and Information Technology, Taif University, P.O. Box 11099, Taif 21944, Saudi Arabia
6 Department of Information Technology, College of Computers and Information Technology, Taif University, P.O. Box 11099, Taif 21944, Saudi Arabia
* Correspondence: mamoon.rashid@vupune.ac.in; Tel.: +91-7814346505

**Abstract:** A Wireless Sensor Network (WSN) is a group of autonomous sensors that are distributed geographically. However, sensor nodes in WSNs are battery-powered, and the energy drainage is a significant issue. The clustering approach holds an imperative part in boosting the lifespan of WSNs. This approach gathers the sensors into clusters and selects the cluster heads (CHs). CHs accumulate the information from the cluster members and transfer the data to the base station (BS). Yet, the most challenging task is to select the optimal CHs and thereby to enhance the network lifetime. This article introduces an optimal cluster head selection framework using hybrid opposition-based learning with the gray wolf optimization algorithm. The hybrid technique dynamically trades off between the exploitation and exploration search strategies in selecting the best CHs. In addition, the four different metrics such as energy consumption, minimal distance, node centrality and node degree are utilized. This proposed selection mechanism enhances the network efficiency by selecting the optimal CHs. In addition, the proposed algorithm is experimented on MATLAB (2018a) and validated by different performance metrics such as energy, alive nodes, BS position, and packet delivery ratio. The obtained results of the proposed algorithm exhibit better outcome in terms of more alive nodes per round, maximum number of packets delivery to the BS, improved residual energy and enhanced lifetime. At last, the proposed algorithm has achieved a better lifetime of ≈20%, ≈30% and ≈45% compared to grey wolf optimization (GWO), Artificial bee colony (ABC) and Low-energy adaptive clustering hierarchy (LEACH) techniques.

**Keywords:** clustering approach; wireless sensor networks; oppositional based learning; gray wolf optimization algorithm; network lifetime

## 1. Introduction

Wireless Sensor Networks (WSNs) cope with a collection of sensing devices scattered in the deployed area to sense the physical activities of its surroundings. These sensors utilize an analog–digital convertor (ADC) to gather the data [1]. The collected information will process to the controller or base station (BS). The data received in the BS will process into decisions for several actions in different applications [2]. Several applications use WSNs for weather prediction, dense domain, the medical field, and commercial and industrial

purposes. Generally, sensor nodes are expensive, and they hold the capacity for sensing, processing, and communicating information [3,4]. The detectors in WSN are firm and hold a fragile-sized battery as their energy source. However, replacing or exchanging the energy source is quite complex due to the placement of sensors in complex or non-man movement environments. The WSN suffers from several issues such as scalability, fault tolerance, energy constraints, path establishment, etc. Most of the sensors will drain their energy due to two cases (a) based on data gathering (sensing) and (b) communicating data to BS through hop nodes. Directly transmitting data to BS consumes more energy than sensing its environment and processing data. Furthermore, more sensor energy consumption will decrease the network's lifespan [5].

Moreover, an optimal energy handling model in WSN will increase the network's lifespan and improve its WSN performance. Thus, WSNs use clustering to reduce redundant energy utilization and ameliorate the network's steadiness. In clustering, each cluster group will elect a leader, known as a cluster head (CH), with privileges to communicate with other CHs in the network. In addition, straight information exchange to BS consumes high energy [6–8]. Hence, several researchers proposed an efficient routing protocol to detect the optimal path amid the CHs and BS to decrease energy utilization. Several works are carried out in the literature to determine the optimal CHs in WSN. In WSN, the LEACH protocol is proposed to handle the CHs selection problem. In LEACH, the CHs are processed with the help of the best-fit method, and the rest of the sensors are referred to as cluster members. Furthermore, a sensor must be a member of any one of the CHs. The CHs gather the information composed by the members and communicate the vital information to the BS via one-hop or many-hop modes [9].

Generally, the researchers classified the WSN clustering approaches into centralized, distributed and hybridized. In addition, the researchers classified similar techniques such as LEACH, HEED [10], etc., and unequal techniques such as ULEACH [11], EDUC [12], EEUC [13], etc. Despite that, they categorized the WSN network into two networks: a homogenous network consisting of equal energy for all sensor nodes and a heterogeneous network that has unequal power for all sensor nodes. Recently, several researchers utilized metaheuristic algorithms to tackle the issue of cluster head selection in WSN. Some recent studies deliberate that using meta-heuristic algorithms provides better efficacy than traditional algorithms. Some of the famous algorithms such as genetic algorithm [14], artificial bee colony algorithm [15], gray wolf optimization algorithm [16], bat algorithm [17], firefly algorithm [18], particle swarm optimization algorithm [19], glow swarm optimization algorithm [20], Harris hawk optimizer [21], cuckoo search (CS) [22], gravitational search algorithm (GSA) [23], memetic algorithm [24], etc. are used in solving various optimization problems.

This work focuses on the clustering mechanism in WSNs based on optimization algorithms. Recent works on clustering techniques viz., classical approaches, metaheuristic approaches, and hybrid approaches are extensively examined to reveal the methodology and properties of existing algorithms. Furthermore, the author introduced an OGWO algorithm that merges oppositional-based learning with the generic GWO technique. This proposed methodology enriches the search capability and eradicates the existing algorithm's issue. It identifies the optimal CHs by keeping sight of various parameters in the objective computations. Briefly, in this article, different test phases are processed to ensure the performance of the formulated method.

The main objectives of this work are given below:

- This paper has attempted to introduce an optimal energy-aware CH election methodology for energy-efficient routing in WSN using a novel "hybrid" technique.
- For the optimal selection of CH, the author formulated the fitness function with constraints such as energy consumption, minimal region among the nodes, the workload of elected CHs, and minimal delay during communication.

- Furthermore, the proposed OGWO technique collaborates with the opposition-based learning technique and generic GWO algorithm that dynamically trades off between the exploration and exploitation search processes during the CH electing process.
- Finally, the outcome of OGWO is compared with the existing algorithms such as LEACH, ABC, and GWO under several test cases, which validates the performance of the work.

The organization of this work is illustrated as follows: Section 2 describes the related work of existing clustering approaches in three different aspects. Section 3 discusses the cluster head election framework and the objective function formulation in WSN. Section 4 provides insight into the proposed methodology for optimal cluster head election. Section 5 presents the experimental set-up, parameter assigning and introduced work result analysis with other contrasted algorithms. Section 6 concludes the work with its outcomes toward the optimal CHs election.

## 2. Related Works

This section deliberates the numerous research studies on selecting cluster heads in the WSN to extend the network lifespan. We have collected a diverse set of articles and divided them into three sectors: classical approaches, metaheuristic approaches and hybrid metaheuristic approaches for well-organized cluster head selection in the WSN. The visual representation of generic clustering is given in Figure 1.

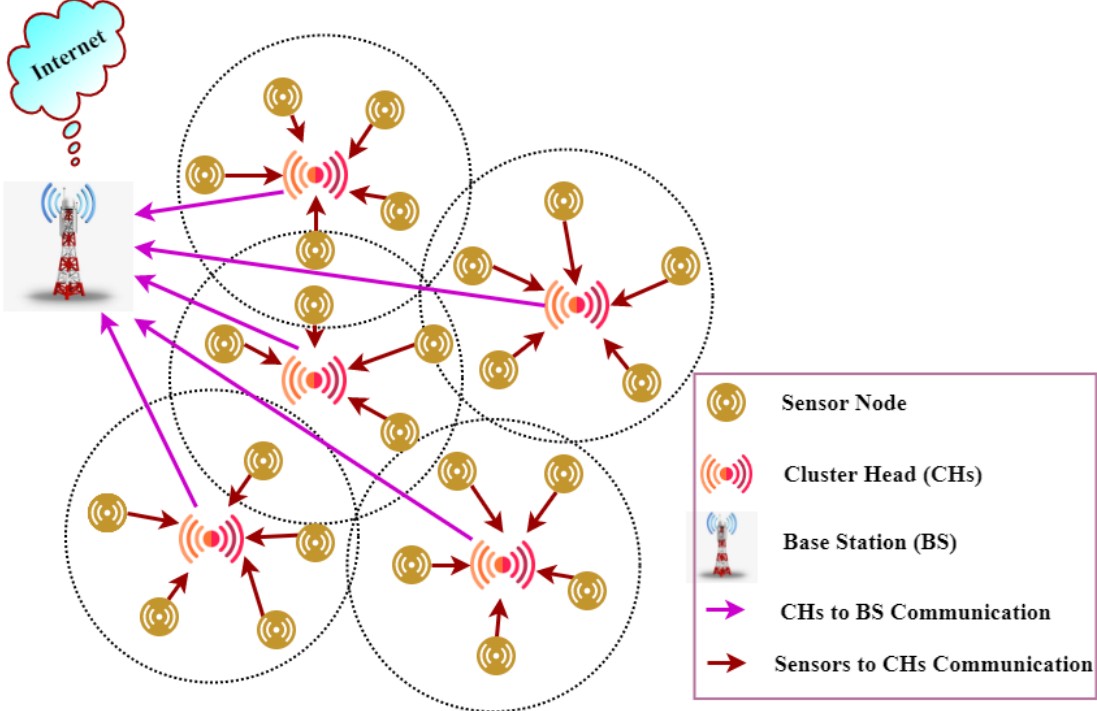

**Figure 1.** Generic Clustering Architecture in WSN.

### 2.1. Classical Approaches

Numerous classical approaches have been introduced to solve the clustering issues, sinking the high energy utilization in the network. LEACH [9] is considered one of the vital algorithms in solving clustering issues of WSN. The LEACH protocol enriches the network lifespan by minimizing the number of packet transmissions by forming the clusters. However, it has several limitations, such as the arbitrary selection of CHs without looking at the distance to BS or remaining energy of the sensors. In addition, another issue is transmitting packets to BS by CH using a single-hop mechanism, which leads the LEACH to suffer in large networks. In LEACH, a set of sensor nodes are grouped into clusters. Researchers have introduced several variations of LEACH approaches to tackle

the above issues. Some variations of LEACH protocols in homogeneous networks include VH-LEACH, LEACH-F, LEACH-C, A-LEACH, O-LEACH, MH-LEACH, IMHT-LEACH, and DMHT-LEACH, which are cast off to elect the CH by considering the remaining power of the sensors.

In [25], the researchers introduced an improved clustering protocol, VH-LEACH. In VH-LEACH, the cluster and CHs have formed arbitrarily. Later, the vice-CHs are nominated by considering the maximum remaining power of elected CHs. However, the performance of the vice-CHs mechanism suffers when there are substantial cluster members. The author in [26] proposed the LEACH-C technique, which works according to the centralized algorithm. In this technique, the BS node makes the clusters and selects the CHs concerning nodes' position and the remaining energy of nodes. The F-LEACH [27] protocol is introduced to address the clustering issue by reducing the delay of the set-up stage and providing efficient CHs distribution. However, the clusters are fixed at the initial state and will be retained for the entire process with no possibility for re-clustering.

Advanced LEACH (A-LEACH) [28] is introduced to eradicate the issue of the LEACH mechanism that reduces energy utilization by electing adequate CHs. A-LEACH intends to enhance the network's lifespan or increase the network's stability for longer epochs by minimizing node death with the aid of heterogeneity attribute parameters. A-LEACH minimizes the data quantity (i.e., data to be transmitted to BS) using the data fusion technique. In addition, A-LEACH selects the nearest gateway node to minimize the data transmission distance. However, A-LEACH selects the CHs arbitrarily and utilizes the single hop for data transmission, leading the technique to provide poor performance in a certain number of iterations. In [29], the Orphan node-based LEACH protocol, namely O-LEACH, is proposed to enrich the better coverage in the network. However, the O-LEACH technique selects the CHs randomly and consumes high energy for grouping the data of neighboring CHs.

In the MHT-LEACH technique [30], the cluster development and head selection are alike to the LEACH initialization process. This technique will not transfer the data directly to the controller; instead, it splits the cluster groups into two divisions, external and internal groups, concerning the location among the sink and CHs. Internal and external groups CHs are selected based on d_0. If the distance of CHs to sink is less than the d_0, then it belongs to internal groups; otherwise, the CHs belong to external groups. Furthermore, DMHT-LEACH [31] and IMHT LEACH [32] are introduced to elect the CHs by considering their residual energy and an equal number of nodes in the cluster. However, the number of cluster heads will vary from one epoch to another, which may decrease the span of the network. In addition, the improved LEACH hierarchal protocols-based information exchange mechanism is discussed in Table 1.

**Table 1.** General Characteristics of LEACH Protocols.

| Protocol | Objectives | Network Type | Parameters | Complexity | Limitations |
|---|---|---|---|---|---|
| VH-LEACH [25] | • To elect the CHs based on residual energy | Homogenous | • Residual Energy | Yes | • Utilizes a single hop to transmit the data from CHs to BS <br> • Additional dealing for VH node |
| LEACH-F [27] | • Utilize centralization for efficient CHs Distribution <br> • Decrease the delay in the set-up process | Homogeneous | • Residual energy | Yes | • At the initial stage, clusters are fixed <br> • No re-clustering processes <br> • Removing the sensor from groups is impossible |
| A-LEACH [28] | • Improve the network stability <br> • Reduce the number of dead nodes | Homogeneous | • Residual energy | Yes | • Arbitrary selection of CHs <br> • Supplementary treatment of CAG nodes |

**Table 1.** *Cont.*

| Protocol | Objectives | Network Type | Parameters | Complexity | Limitations |
|---|---|---|---|---|---|
| O-LEACH [29] | • Better coverage of network <br> • Orphan node election to transmit the data | Homogeneous | • Residual energy <br> • Distance | Yes | • Arbitrary selection of CHs <br> • Single Hop communication is utilized |
| MHT-LEACH [30] | • Multi-hop technique <br> • Division of CHs into two groups | Homogeneous | • Distance <br> • Residual Energy | Yes | • Selects CHs without considering node energy <br> • No. of the cluster are not equal |
| DMHT-LEACH [31] | • Multi-hop technique <br> • Division of CHs into multi-groups based on distance and energy | Homogeneous | • Distance <br> • Residual Energy | Yes | • Arbitrary selection of CHs <br> • No. of the cluster are not equal |
| IMHT-LEACH [32] | • Multi-hop technique <br> • Division of CHs into multi-groups | Homogeneous | • Distance <br> • Remaining Energy | Yes | • Random CHs election <br> • The distance among CHs-to-CHs members is not considered |
| TB-LEACH [33] | • To improve the network lifespan | Homogeneous | • Distance <br> • Residual Energy | Yes | • It depends on the random timer <br> • No. of the cluster is fixed for all epochs |
| I-LEACH [34] | • To elect the CHs based on distance and power | Homogeneous | • Distance <br> • Residual Energy | No | • Not considered node centrality |
| BRE-LEACH [35] | • To elect the CHs based on remaining energy and distance among the node to BS | Homogeneous | • Remaining Energy <br> • Distance | Yes | • Node with maximum energy only considered for CH selection |
| EADCR-LEACH [36] | • To enhance the system lifespan <br> • To elect the CHs based on distance and remaining energy | Homogeneous | • Residual energy <br> • Distance | Yes | • Not considered node centrality |

### 2.2. Metaheuristic Approaches

In this sub-section, we discussed recent metaheuristic algorithms utilized to solve the clustering issues in WSN. Generally, these algorithms are classified into two major divisions: evolutionary algorithms and swarm intelligence algorithms. The main aim of developing such algorithm is to handle the NP-hard glitches, which classical approaches may not translate in a stipulated period [37]. Although the algorithm suffers several challenges in obtaining optimal solutions, merging clustering approaches to metaheuristic algorithms attains better performance in minimizing the energy consumption in WSNs. Based on the benefits of these techniques, a wide range of researchers have utilized several metaheuristic algorithms to solve the clustering issues in WSN [38].

In [39], the author utilized an evolutionary-based algorithm, namely a genetic algorithm (GA), for solving clustering and routing issues in WSNs. The GA enhances the CHs lifetime to prolong the network lifespan. However, generic GA suffers from local optimal errors that might lead to poor performance during iterations. To eradicate the issue, the author in [40] introduced a GA-based threshold-subtle energy-efficient cluster selection mechanism that uses cohesion and cluster division processes. The author utilized the inter-cluster data communication technique to extend the system lifespan by considering the nodes' load balance and the nodes' residual energy. The author introduced the multipath routing protocol [41] by hybridizing the active bunching and ant colony optimization (ACO) algorithm. The algorithm uses three segments to elect the CHs and route among the cluster

members from CHs to BS. This three-phase process aids the network in prolonging the lifetime by selecting optimal CHs. However, the stability of the network path is inefficient in the course of iterations.

The author of [42] introduced an energy-efficient clustering algorithm with the aid of a swarm intelligence-based artificial bee colony (ABC-SD). This technique reduces energy consumption by intensifying the ABC's search process. Furthermore, the centralized control technique simulates the LP formulation that handles the multi-objective function within the sink node. The author in [43] proposes the Fractional Lion (FLION) clustering technique. This clustering technique includes the enduring power of the sensor node and the distance between the CHs to BS to elect the CHs. The cluster formation is processed based on objectives such as inter and intra-cluster space, residual energy and delay. The algorithm's performance is associated with other mechanisms such as PSO, LEACH, and ABC, and the Fractional ABC technique showed that the protocol enriches the QoS metrics.

In ref [44], the author introduced two-tier PSO for handling the clump and routing process (TPSO-CR). TPSO-CR protocol is used to mitigate the clustering issues by electing the optimal CH in view of the remaining energy and distance among the nodes. The author [45,46] introduced a modified GSA algorithm to determine the optimal base station location in two-tiered heterogenous WSNs.

In addition, we have discussed some sets of other metaheuristics-based clustering approaches in Table 2.

**Table 2.** Review of metaheuristics-based clustering approaches.

| Algorithm | Year | Objectives | Mechanism | Metrics | Complexity | Simulation |
|---|---|---|---|---|---|---|
| ICWAQ [15] | 2012 | Reduce energy consumption | • ICWAQ intensifies the better and more efficient ABC technique to optimize senor clustering | • Residual energy<br>• Throughput | Yes | MATLAB |
| HACH [47] | 2017 | Network lifetime | • GA-based method to move actively to inactive nodes | • Average energy<br>• Stability<br>• Network lifetime | Low | MATLAB |
| EC-PSO [48] | 2019 | Energy hole | • Geometric-based CH election<br>• Nodes close to the energy center are elected using improved PSO | • Average energy consumption<br>• The average number of hops<br>• Alive node | Yes | MATLAB |
| I-FBECS [49] | 2021 | Network lifetime | • Novel fitness function is formulated<br>• The rank-based technique is used for non-cluster nodes | • Alive nodes per round<br>• First node death<br>• Half node death<br>• Average energy consumption<br>• Throughput | Yes | MATLAB |
| LB-CR-ACO [50] | 2018 | Network lifetime | • Priority weights are assigned to elect the CHs<br>• Dynamic selection of CHs in every epoch | • Average energy<br>• Throughput<br>• Packet delivery ratio | Yes | MATLAB |
| MHACO-UC [51] | 2019 | Reduce energy consumption | • Electing relay nodes to decrease the maximal distance of data transmission<br>• Link maintenance and neighbor finding using MHACO-UC | • Packet delivery ratio<br>• Energy consumption<br>• Residual energy<br>• Node death rate | Yes | MATLAB |
| GWO-CH [52] | 2020 | Network lifetime | • GWO algorithm used to select the optimal CHs<br>• Mitigate the energy holes | • Energy consumption<br>• Residual energy<br>• Node death rate | Low | MATLAB |
| SMO-CH [53] | 2018 | Load balancing Network lifetime | • Threshold-sensitive energy-efficient protocol to elect the CHs | • Energy consumption<br>• Network lifetime | Yes | MATLAB |

### 2.3. Hybrid Metaheuristic Approaches

The author [54] addressed cluster head selection and sink mobility-based data communication by introducing a hybrid GAPSO algorithm. This algorithm merges GA and PSO to improve the system lifespan. The author [55] formulated a hybrid procedure named dolphin echolocation-based crow search technique to handle the clustering problem. In this algorithm, CHS is elected by considering multiple constraints and improves the convergence rate. Furthermore, the energy-aware channeling is processed in data for efficient data transmission. The algorithm's performance is validated using different node scenarios, achieving a better network lifetime. The work in [56] introduced a modified hybrid algorithm, namely artificial bee colony and firefly algorithm (HMABCFA), to elect the optimal cluster head. The author improved the exploration and exploitation process in the standard ABC and FA algorithm and achieved a trade-off between search processes. The performance of the HMABCFA improved the network's lifetime and energy stability and decreased the network latency compared to other approaches.

The author in [57] proposed a novel fitness function and a hybrid glowworm swarm with fruitfly algorithm (FGF) to elect the optimal CHs. The optimal CHs improved the network's lifetime with respect to parameters such as delay, distance and residual energy in the CH election. A hybrid algorithm was introduced in [58], namely the harmony search algorithm and PSO (HSAPSO), in which intensification and exploitation are improved to hand pick the optimal CHs. Yet, the author did not consider the significant parameters such as node centrality and node degree to elect the cluster head, which might decrease the network's performance. Later, a hybrid technique, namely the firefly technique with particle swarm optimization (HFAPSO) [59], is introduced to determine the optimal CHs in the LEACH-C approach. The performance of the HFAPSO technique is evaluated based on the count of alive sensors, remaining energy, dead sensors and throughput. The resulting outcome was that the introduced technique enhanced the network's lifetime. In addition, hybrid gray wolf and crow search optimization (GWCSO) is introduced to select the cluster head by considering minimal delay, minimum distance, and energy stabilization [60].

### 2.4. Exact from the Literature

The main drawbacks of the previous studies in the literature are:

(i)     The capability of maintaining the trade-off in the search method is not sustained and fails to obtain the optimal solution within a minimal time.

(ii)    The energy trade-off guaranteed by the previous metaheuristic techniques is inadequate to improve or sustain the network lifespan.

The actual literature work motivates us to propose a novel hybrid optimization algorithm, namely the gray wolf optimization algorithm with oppositional-based learning, to determine the optimal CH. It also sustains the trade-off between the exploration and exploitation processes.

## 3. Energy-Aware Cluster Head Selection Framework

This section elaborates on the concept of the network, energy, distance, and objective models used for experimental purposes in detail. Furthermore, we also discussed the network's lifespan and parameters used in this work.

### 3.1. Network Model

In this work, WSN consists of $n$ several collective sensors and a BS. Furthermore, the network model is adopted from ref. [61], and the set-up of the WSN has the following parameters:

(a)     All sensor nodes in WSN are arbitrarily scattered among the 2D plane of the sensing environment that includes unique latitude and longitude location points.

(b)     Sensor nodes are energy constrained; once the sensors are deployed in the sensing environment, they are left unattended, since recharging them is unrealistic.

(c)    All the sensors are consistent and hold typical processing and transmission proficiencies; thus, they utilize the equal energy level for the transmission and processing of data bits.

(d)    Once the sensors are deployed in the sensing field, they are static concerning BS; all sensors in the network have equal opportunities to act as a regular node or CH.

(e)    All sensor nodes should detect data about their current circumstance and the same to be communicated to CH. Furthermore, the number of sensor nodes should be more prominent than the number of CHs.

(f)    The position of the BS is changeable according to the analysis of performance within the sensing region.

(g)    The transmission route between the sensor nodes and CHs is wireless, and its path is determined within the transmission region.

(h)    Finally, the sensor nodes can avail different communication power hierarchies concerning data transmission distance.

### 3.2. Energy Utilization Model

We took on the energy use model based on the author's reference in [61]. In this model, we processed the overall network energy utilization ($E$) in light of the energy dispersed by the transmitter ($E_{TX}$) and receiver ($E_{RX}$), and we numerically figured it out as below:

$$E_{Total}(n,\theta) = E_{TX}(n,\theta) + E_{RX}(n) \tag{1}$$

where $E_{Total}(n,\theta)$ represents the overall network energy consumption, and $E_{TX}(n,\theta)$ denotes the energy utilized to operate the radio amplifier and power electronics. The mathematical formulation of energy consumption by the transmitter for communicating $n$ bits of information is given by:

$$E_{TX}(n,\theta) = \begin{cases} n \times E_{elec} + n \times \varepsilon_{fs} \times \theta^2 & if\ \theta < \varphi \\ n \times E_{elec} + n \times \varepsilon_{mp} \times \theta^4 & if\ \theta \geq \varphi \end{cases} \tag{2}$$

where $E_{elec}$ denotes the energy consumed per bit to run the transmitter. $\varepsilon_{fs}$ and $\varepsilon_{mp}$ denote the amplification energy for the free space model and multipath model, whereas $\varphi$ represents the threshold communicating distance, and its rate is measured by $\varphi = \sqrt{\frac{\varepsilon_{fs}}{\varepsilon_{mp}}}$. $\theta$ denotes the distance parameter for computing transmitter energy utilization concerning the volume of information exchange. If the information exchange is in the $\varphi$, then the transmittance energy is equal to $\theta^2$; otherwise, it is $\theta^4$. Therefore, the distance and workload are considered significant parameters to improve the network lifetime.

Further, energy utilization by the beneficiary sensor for handling $n$-bits of data ($E_{RX}(n)$) is given by:

$$E_{RX}(n) = n \times E_{elec} \tag{3}$$

The overall lifetime of the network ($NL$) is calculated with respect to residual energy level termed as ($E_{residual}$) and the total energy of the sensor termed as ($E_{total}$) after exchanging; the $n$-bit information, which is articulated as below:

$$NL\big(S_i,\ CH_j\big) = \frac{E_{residual}^i}{E_{total}^i(n,\theta)} \tag{4}$$

where $NL\big(S_i,\ CH_j\big)$ denotes the network lifetime concerning $i$ number of sensor nodes (i.e., $S_i \in SN$) and $j$ number of cluster heads elected; $E_{residual}$ represents the remaining energy of the sensors, and $E_{total}$ represents the total energy expended by the sensors. We computed the network lifetime concerning the first node dead (FND).

### 3.3. Distance Model

Generally, any communication among the sensor nodes to CH or CH to BS may require some amount of energy according to the role or position acted by the node in the network. The communication of information between the sensor with the maximum distance might consume high energy, whereas the information of data with less space consumes less power. We computed the distance among the sensor nodes to BS as:

$$\theta_i = \sqrt{(x_{BS} - x_i)^2 - (y_{BS} - y_i)^2} \; ; \; (i = 1, 2, \ldots, SN) \tag{5}$$

where $\theta_i$ denotes the distance of the $i$th sensor node to BS position; $(x_{BS}, y_{BS})$ represents the x- and y-coordinates of the BS; $(x_i, y_i)$ specifies the position of the $i$th sensor node; $SN$ denotes the number of sensors distributed in the sensing region.

Further, the Euclidean distance between the sensor and CH is computed as follows:

$$\theta\left(SN_i, N_{CH_j}\right) = \sqrt{(x_j - x_i)^2 - (y_j - y_i)^2} \; ; \; (i = 1, 2, \ldots, SN; j = 1, 2, \ldots, N_{CH}) \tag{6}$$

where $N_{CH}$ denotes the number of cluster heads elected so far.

### 3.4. Objective Model

This subsection formulated the fitness function for electing the optimal CH among the typical sensors. The formulation of the fitness function utilizes the five different parameters such as the sensors' remaining energy, distance model (i.e., the distance among sensors, CHs and BS), node degree and node centrality.

(a)    *The residual energy of the CH*

Initially, we use the residual energy of the sensor node to eradicate the non-alive nodes as a CH in the clustering procedure. CH performs various assignments such as collecting information from other sensors (i.e., cluster members), accumulating the information, and communicating the information to BS. Thus, the CH requires high energy to perform the above-mentioned assignments. So, we prioritized the sensor with maximum residual liveliness to act as CH. The residual energy ($f_1$) is illustrated as follows:

$$f_1 = \sum_{j=1}^{N_{CH}} \frac{1}{E_{CH_i}} \tag{7}$$

where $E_{CH_i}$ denotes the remaining energy of the $i$th sensor node.

(b)    *The distance among the sensor nodes*

Secondly, we compute the distance among the cluster members and their CH. The senor node energy overindulgence is due to the length of the transmission path, as stated in Section 3.2. The energy utilization is high when the transmission distance is more and vice versa. We mathematically formulated the interval among the typical sensor and CH ($f_2$) as:

$$f_2 = \sum_{i=1}^{SN} \left( \sum_{j=1}^{N_{CH}} \theta\left(SN_i, N_{CH_j}\right) / N_{CH} \right) \tag{8}$$

where the interval among sensor $i$ and $N_{CH_j}$ is represented as $\theta\left(SN_i, N_{CH_j}\right)$.

(c)    *Distance between CH and BS*

It stipulates the interval between the CH and BS. The sensor energy mainly relies on the length of the communication track. For instance, let us consider that BS is distant from the CH; then, it entails high energy for information exchange. Hence, the abrupt changes in CH energy levels are due to excess energy utilization. Therefore, the node with minimal

distance to BS is given higher priority for information exchange. We mathematically formulated the fitness function ($f_3$) of the distance among the CH and BS as:

$$f_3 = \sum_{j=1}^{N_{CH}} \theta\left(N_{CH_j},\ BS\right) \tag{9}$$

where the distance among the $N_{CH_j}$ and Bs is represented as $\theta\left(N_{CH_j},\ BS\right)$.

*(d)   Node degree*

It represents the collection of sensors grouped to the corresponding CH. Due to energy constraints, we elected the CH with a limited number of sensors. The CH with a high number of cluster members requires high energy for data collection and aggregation; therefore, it will reduce the lifespan of CH over time. We formulated the node degree ($f_4$) as:

$$f_4 = \sum_{j=1}^{N_{CH}} N_{CH_j} \tag{10}$$

where $N_{CH_j}$ is denoted as the number of $j$ cluster heads.

*(e)   Node centrality*

It represents the number of neighbor nodes surrounded by a sensor node or the node which is centrally positioned from the adjacent nodes, and we mathematically expressed it as:

$$f_5 = \sum_{j=1}^{N_{CH}} \frac{\sqrt{\sum_{i \in m} \theta^2(j,\ i)/m(i)}}{Network\ area} \tag{11}$$

where $m(i)$ is denoted as the number of adjacent nodes of $N_{CHj}$.

We converted the multi-objective function into a single-objective process using weight factors for each fitness function. The weight factors were $\vartheta_1$, $\vartheta_2$, $\vartheta_3$, $\vartheta_4$ and $\vartheta_5$. We formulated the overall objective function as given below:

$$f = \vartheta_1 f_1 + \vartheta_2 f_2 + \vartheta_3 f_3 + \vartheta_4 f_4 + \vartheta_5 f_5 \tag{12}$$

where the factors of $\vartheta_1$, $\vartheta_2$, $\vartheta_3$, $\vartheta_4$ and $\vartheta_5$ are assigned the value of 0.3, 0.25, 0.2, 0.15 and 0.1, respectively. Firstly, the weight factor $\vartheta_1$ is considered a high priority because of residual energy of CH, which may eradicate the electing node with less energy as a CH. Then, the second and third superiority weight factors $\vartheta_2$ and $\vartheta_3$ are used to determine the interval among the typical sensor to CH to BS. Later, the weight factor $\vartheta_4$ is considered the fourth superiority for electing CH with a minor node degree. Finally, the weight factor $\vartheta_5$ is assigned as the fifth priority that aids in improving the closeness among the CH and corresponding cluster members.

## 4. Proposed Methodology

This section discusses the solution representation of OGWO methodology, conventional GWO process and limitations, and oppositional-based learning techniques. In addition, the working methodology of the proposed work in determining the cluster head selection is presented.

### 4.1. Solution Representation

This work has introduced a hybrid optimization algorithm, namely OGWO, which merges the conventional GWO and oppositional-based learning algorithm to elect the energy-aware optimal CH within the network. We formulated the solution representation for the proposed algorithm as shown in Figure 2, in which $\left(CH_1,\ CH_2,\ \ldots,\ CH_{N_{CH_j}}\right)$ is the CHs and $N_{CH_j}$ represents the total number of cluster heads.

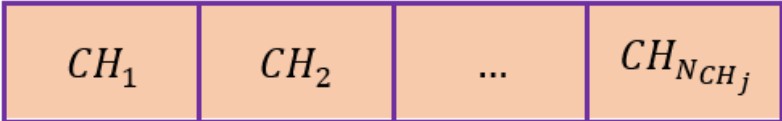

**Figure 2.** Solution Representation.

*4.2. Conventional Gray Wolf Optimization*

Seyedali Mirjalili recently introduced gray wolf optimization (GWO) in 2014 [62], in which the intellectual behaviors, namely the good leadership and hunting strategy of gray wolves, are represented. Generally, gray wolves hunt the prey based on the group-based hunting mechanism that includes a pack of 5–12 wolves gathering together to attack the target. The collection of wolves works in a four-level leadership hierarchy; namely, the first leader termed alpha ($\alpha$), the second leader denoted as beta ($\beta$), the third leader termed delta ($\delta$), and the rest as members termed omega ($\omega$). The $\alpha, \beta,$ and $\delta$ are dominant wolves which control the ($\omega$) to sustain the safety and integrity of the pack. The author mathematically formulated the working process of gray wolves in three methods: encircling, hunting and searching.

(a)    *Encircling*

Initially, gray wolves process the encircling to trap the prey before initiating the hunting process. The encircling method is expressed mathematically as below:

$$\vec{Y} = \left| \vec{C}. \vec{P_p}(mx) - \vec{P}(mx) \right| \tag{13}$$

$$\vec{P}(mx+1) = \left| \vec{P_p}(mx) - \vec{A}.\vec{D} \right| \tag{14}$$

where $\vec{Y}$ denotes the interspace between the wolf and the prey, $\vec{P}$ specifies the present location of the wolf in $mx$ generations and $\vec{P_p}$ determines the prey location. The coefficient parameters, namely $\vec{A}$ and $\vec{C}$, are computed as below:

$$\vec{A} = 2\vec{a}.\vec{Y_1} - \vec{a} \tag{15}$$

$$\vec{C} = 2.\vec{Y_2} \tag{16}$$

where $\vec{Y_1}$ and $\vec{Y_2}$ specify the random values computed within the boundary of [0, 1]. These values help to change the circumference of wolves randomly toward the prey. The parameter $\vec{a}$ is used to limit the crusade of the technique, which slowly converges within the range of [2, 0].

(b)    *Hunting*

Secondly, the hunting process is initiated slowly by adjusting the location of all the $\omega$ wolves with the aid of dominant wolves $\alpha, \beta,$ and $\delta$. The author mathematically formulated the location adjustment of dominant wolves as:

$$\vec{Y_\alpha} = \left| \vec{C_1}. \vec{P_\alpha} - \vec{P} \right|, \vec{Y_\beta} = \left| \vec{C_2}. \vec{P_\beta} - \vec{P} \right|, \vec{Y_\delta} = \left| \vec{C_3}. \vec{P_\delta} - \vec{P} \right| \tag{17}$$

$$\vec{P_1} = \left| \vec{P_\alpha} - \vec{A_1}. \vec{Y_\alpha} \right|, \vec{P_2} = \left| \vec{P_\beta} - \vec{A_2}. \vec{Y_\beta} \right|, \vec{P_3} = \left| \vec{P_\beta} - \vec{A_3}. \vec{Y_\beta} \right| \tag{18}$$

The author formulated the overall position update of all wolves using Equations (17) and (18) as:

$$\vec{P}(k+1) = 0.33 * \sum_{i=1}^{3} \vec{P_i} \tag{19}$$

where $\vec{P_i}$ denotes the arbitrary position of wolves concerning the distance between the $\alpha, \beta$, and $\delta$ wolves.

(c)    *Attack and search the prey*

Finally, the attack and search prey define the prey attack by the wolf and the search for a new target within the search boundary. The coefficient parameter $\vec{A}$ generates the random value to intensify and diversify the search location of the gray wolves. Gray wolves strengthen the spot toward the prey if $\left|\vec{A}\right| < 1$, or else they search for a new target or prey (i.e., $\left|\vec{A}\right| > 1$). The parameter $\vec{C}$ linearly adjusts its values within the limit of [0, 2], which prevents the technique from internal stagnation.

The author formulated the working principle of generic gray wolf optimization as given in Algorithm 1.

---

**Algorithm 1**. Generic Gray Wolf Optimization.

---

1: Set the parameters such as population size, A and C
2: Generate the random position of wolves $P_i$ within the search region
3: Compute the fitness of wolves $f_i$
4: Determine the $\alpha, \beta$, and $\delta$ dominant wolves
5: While ($mx \leq$ max_*Iter*) // Initially, $mx = 1$
6:    For $i = 1 : N_p$
7:        Modify the wolf position using Equation (19)
8:        Compute the fitness of wolves $f_i$
9:    End for
10: Update the $\alpha, \beta$, and $\delta$ dominant wolves
11: Increase $mx$ value to 1 for every iteration (i.e., $mx+ = 1$)
12: End while

---

*4.3. Opposition-Based Learning Technique*

The opposition-based learning technique (OBL) was formulated by Tizhoosh [63] to boost the convergence of the traditional metaheuristic algorithms. This method utilizes the valuation of the contemporary population rather than the opposite population to determine a better solution for a specific problem. The OBL method has been used in different metaheuristic algorithms to boost the convergence speed. The mathematical model of the OBL is defined as below:

Let $\mu(\mu \in [p, q])$ be an actual integer. The contradictory integer $\mu^0$ is formulated as

$$\mu^0 = p + q - \mu^0 \tag{20}$$

For $d$—dimensional pursuit space, the contradictory integer $\mu^0$ is defined as

$$\mu_j^0 = p_j + q_j - \mu_j \tag{21}$$

where $\mu_1, \mu_2, \ldots, \mu_D$ is a theme in d-dimensional pursuit space and $\mu_i \in [p_j, q_j]$; $j = \{1, 2, 3, \ldots, d\}$.

This oppositional-based technique is utilized at the time of initialization procedure and also in every generation with the aid of iteration jumping rate $J_r$. The $J_r$ parameter is used to explore the search space and eradicate the local optimal struck. The author represented the process of OBL as given in Algorithm 2.

---

**Algorithm 2**. Oppositional-Based Learning Algorithm.

---

1: Foremost, the algorithm initializes random solutions with the upper and lower boundary regions
2: Determine the opposite solutions:
2.1: *for i* = 1 : $N_p$
2.2: *for j* = 1 : *d*
2.3: $\quad \mu_{i,j}^0 = p_j + q_j - \mu_{i,j}$
2.4: *end for*
2.5: *end for*
3: Sort the current and opposite solutions into minimum to maximum values.
4: Choose $N_p$ number of best candidate solutions from the recent and contrary solutions.
5: Update the control parameters for the quantified problem utilizing the OBL technique.
6: Generate the opposite solutions from current solutions using the jumping rate $J_r$:
6.1: *for j* = 1 : $N_p$
6.2: $\quad$ *for i* = 1 : *d*
6.3: $\quad$ *if* $J_r$ > *rand*
6.4: $\quad\quad$ *opp(j, i)* = *min(i)* + *max(i)* − *P(j, i)*;
6.5: $\quad$ *else*
6.6: $\quad\quad$ *opp(j, i)* = *P(j, i)*;
6.7: $\quad$ *end*
6.8: $\quad$ *end for*
6.9: *end for*
7: Sort the solutions (*P*) and opposite solutions (opp) from minimum to maximum and choose the $N_p$ number of best candidate individuals from the recent and opposite solutions.
8: Replicate step 5 until the end criterion is satisfied.

---

### 4.4. Proposed Algorithm

OBL is a recent concept in machine learning that mimics the process of opposite relationships between entities. Researchers have widely used this algorithm to enhance the convergence speed and boost metaheuristic algorithms' search processes. GWO is a variant of the swarm intelligence family that mimics the working principle of the gray wolf that intakes the leadership and hunting strategy. This algorithm has inspired several researchers with its simplicity and ease of use in solving several complex optimization problems. However, the conventional algorithm suffers from common issues such as local optimal error and premature convergence. It may lead to poor accuracy in determining optimal solutions in multi-model optimization problems [64]. We hybridized the OGWO algorithm to overcome the issues mentioned above. This hybridized algorithm merges the OBL and the GWO algorithm to progress the search capabilities of GWO and boost the convergence in electing the optimal CHs within the network. The functional architecture of cluster head selection using OGWO is given in Figure 3.

The working process of OGWO is as follows: Firstly, we initialized the population by the OBL method within the search limits in the proposed method. Later, the position of wolves is updated using conventional GWO, and OBL determines the opposite part of wolves. Moreover, the proposed algorithm updates wolves' location by merging the best OBL and GWO algorithms. The algorithm sustains the trade-off among intensification and diversification in searching for the optimal CHs within the network. We formulated the working algorithm of OGWO as given in Algorithm 3.

---

**Algorithm 3**. Cluster Head Selection Using Proposed Technique.

---

1: Generate arbitrary initial population $\Phi$;
2: For $i = 1 : N$
3:     $r_{1,i} = rand(0,1)$, $r_{2,i} = rand(0,1)$;
4:     For $j = 1 : D$
5:         $\Phi_{ij}^{do} = \Phi_{ij} + r_{1,i}.(r_{2,i}.(\lambda_j + \alpha_j - \Phi_{ij}) - \Phi_{ij})$;
6:         Ensure the search boundary;
7:     End For
8: End For
9: Compute the fitness of all search agent
10: Pick the top best N solutions from $\Phi^{do} \cup \Phi$ to $\Phi^S$
11: Determine the first three best search agents of $\alpha$, $\beta$ and $\delta$ from $\Phi^S$
12: While $t <= max\_Iter$
13: For $i = 1 : N$
14:     Modify the search agents' position $\Phi^S$ using Eq. (2)
15:     Ensure the boundary limits of all search agents;
16: End for
17:     For $i = 1 : N$
18:         If $rand < \delta$
19:             $r_{3,i} = rand(0,1)$, $r_{3,i} = rand(0,1)$;
20:         For $j = 1 : D$
21:             $\Phi_{ij}^{do} = \Phi_{ij}^S + r_{1,i}.\left(r_{2,i}.(\lambda_j + \alpha_j - \Phi_{ij}) - \Phi_{ij}^S\right)$;
22:             Ensure the boundary limits;
23:         End for
24:     End if
25: End for
26:     Compute the fitness of all search agents
27:     Pick the top best N solutions from $\Phi^{do} \cup \Phi$ to $\Phi^S$
28:     Update the three best search agents of $\alpha$, $\beta$ and $\delta$ from $\Phi^S$
29: Update the power exponent value
30: End while
30: Output: CHs from the network (optimal solution)

---

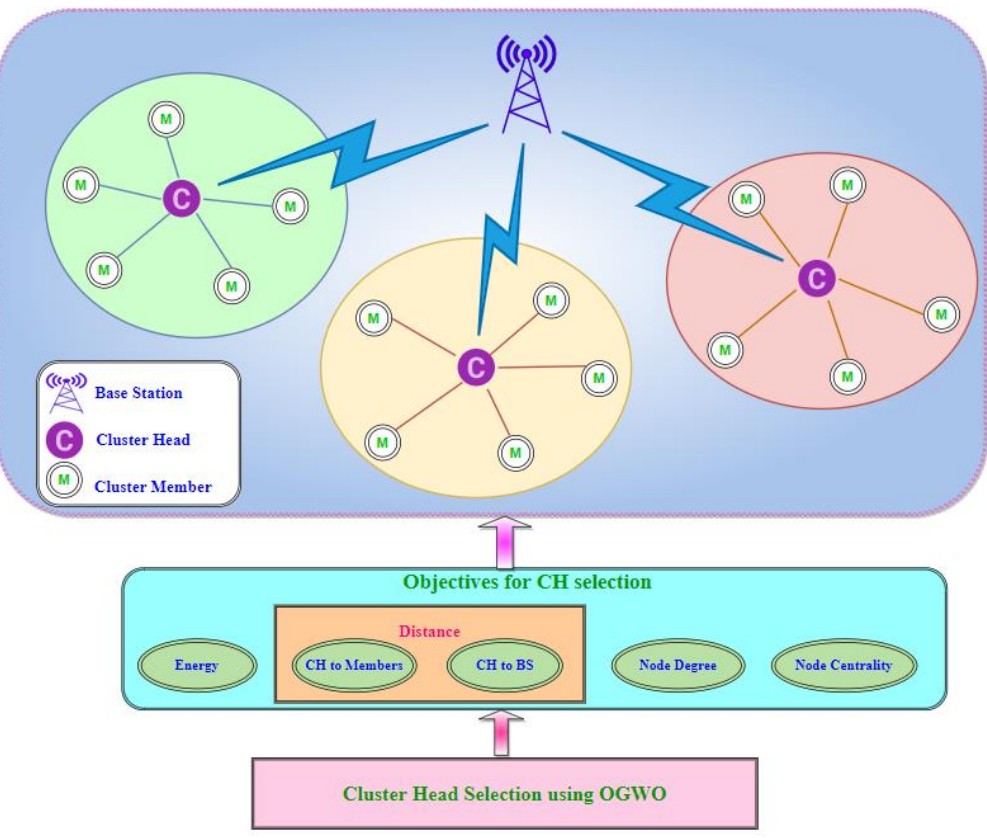

**Figure 3.** Cluster Head Selection Using OGWO.

### 4.5. Exploration and Exploitation Process of Proposed Algorithm

In our proposed work, we incorporated the OBL technique to enhance the search process of traditional GWO in terms of exploration. This OBL technique helps the proposed algorithm explore the search space using jump rate $J_r$. This parameter helps eradicate the local optimal error and overcome the slow convergence rate. Based on the experimentation, we fixed the $J_r$ value as 0.4 that provides better outcome than traditional GWO. In addition, the coefficient parameter $\left|\overrightarrow{A}\right| < 1$ aids the proposed algorithm to exploit the local search space.

## 5. Experimentation and Result Analysis

This section illustrates the experimentation of the proposed methodology, performance metrics, and analysis of proposed work in terms of four test cases with respect to the first node dead, number of alive nodes, packet delivery and impact of BS and CHs.

### 5.1. Experimental Set-Up

In this section, we set up a simulation environment in MATLAB version 2018a, which functioned in the Windows 10 operating system with a hardware platform of Intel Xenon, i5-3570 CPU with a speed of 3.6 GHz and 16 GB RAM, 10 MB cache, respectively. Selecting the MATLAB tool is due to the ease of mathematical operations and adequate data examination. In our work, we randomly scattered 400 sensors in the network within the deployment zone of $200 \times 200$ m$^2$. We presented the experimentation parameters in Table 3. The main goal of this work is to identify the optimal cluster head to improve the network's lifespan. The author of this work used numerous input parameters to elect the finest CHs. Furthermore, we presented the OGWO parameters in Table 4, and the parameters of GWO [52] and ABC [65] were adopted from the authors' reference. We compared the proposed OGWO algorithm with a few state-of-the-art conventional algorithms such as GWO, ABC, and LEACH. The authors of the traditional algorithm proved their efficacy in improving energy efficiency in WSN.

**Table 3.** Network simulation parameters.

| Parameter | Value |
| --- | --- |
| Deployment Area | $200 \times 200$ m$^2$ |
| BS Location | (0,0) (50,50), (100,100), (150,150) |
| Number of Senor Nodes | 100 to 400 Nodes |
| Initial Node Energy | 2.0 J |
| Number of CHs (%) | 10% to 25% |
| $E_{elec}$ | 50 nJ/bit |
| $E_{fs}$ | 10 pJ/bit/m$^2$ |
| $E_{mp}$ | 0.0013 pJ/bit/m$^4$ |
| $D_{max}$ | 100 m |
| $D_o$ | 30 m |

**Table 4.** Parameters of proposed algorithm.

| Parameter | Value |
| --- | --- |
| Number of wolves | 100 |
| Maximum number of Iterations | $10 \times 10^3$ |
| Jumping rate ($\delta$) | 0.4 |
| Coefficient parameter (c) | [2, 0] |

### 5.2. Performance Evaluation Metrics

Here, we discussed the evaluation performance metrics as follows:

*First Node Dead (FND):* It frames the number of cycles until the first sensors dies in the network. We utilized this measurement to decide the maximum duration the network can endure in dynamic mode.

*The Number of Alive Nodes (NoAN)* determines the number of active sensors in the network. The network life expectancy broadens when the number of active sensors is high.

*Number of Packets Received (NoPR)* by BS: The total number of packets received at BS is directly relative to the alive nodes and the remaining sensor energy. If the active nodes are high, the number of packets accepted by BS is high.

*Average energy utilization* specifies the collective power each sensor uses per generation.

### 5.3. Result Analysis and Discussion

In this work, we used diverse test cases for our experimentation to analyze various outcomes. We used a set of numeral sensors (NSNs) in our investigation, and we observed the corresponding results. Furthermore, the effects are marked based on the impact of BS and CHs. We considered four test cases to validate the performance of the formulated method. We discussed the detailed set-up of each test case and its outcomes as follows:

*(i) Test Case 1—FND:* In this test case, we used the deployment area as $200 \times 200$ m$^2$, the NSNs as 100 to 400, the position of BS fixed as (100,100), and the NCHs were 10%, 15%, 20%, and 25%. The FND outcome of the proposed methodology is measured based on the number of rounds (NoR) in the network. Test case 1 deliberates the results of FND over the number of rounds. The performance measure is significant in analyzing the algorithm's efficacy by selecting optimal CHs in the network. Figure 4 presents the FND performance of the proposed methodology and existing compared methods. We used a different set of sensor nodes such as 100, 200, 300, and 400 in this experimentation, and we graphically illustrated the outcome in Figure 4a–d, respectively. A comparison of the proposed work with traditional clustering algorithms such as LEACH, ABC, and GWO is presented here. We noticed that our proposed OGWO algorithm attains a better outcome of ≈50%, ≈30% and ≈20% over LEACH, ABC and GWO, respectively.

The reason behind the achievement of lifetime (LT) in OWGO over LEACH is due to certain limitations of LEACH. Generally, the LEACH algorithm utilizes the probabilistic method and arbitrarily selects CHs, resulting in high energy ingesting and reducing the lifespan of the network. Furthermore, LEACH selects the minimal residual energy sensor as CH. Similarly, the ABC algorithm performs well in the population-based clustering approach, but it fails to consider the workload of the sensor. On the other hand, the conventional GWO algorithm provides a better solution for several complex optimization problems. However, it fails to explore the search space and suffers regarding maintaining the trade-off between intensity and diversity. Compared to conventional GWO, ABC and LEACH techniques, our proposed OGWO algorithm attains more rounds after FND. Furthermore, the OGWO algorithm outperforms better in exploring the search space during cluster formation. Our proposed algorithm achieves a higher network lifespan than LEACH, ABC and GWO algorithms concerning FND over number of rounds.

*(ii) Test case 2—NoANs:* In this test case 2, we considered the sensing region as $200 \times 200$ m$^2$, the NSNs as 100 to 400, the position of BS as (100, 100), and the NCHs fixed as 25%. Here, we examine the performance of the OGWO with respect to the number of alive sensors. The outcome of test case 2 determines the NoANs over a maximum number of rounds. The observed results of the formulated OWGO with LEACH, ABC and GWO are presented in Figure 5. The x and y coordinates of Figure 5 represent NoR and NoANs, respectively. Figure 5a–d present the NoANs comparison with respect to NSNs of 100, 200, 300 and 400 sensors. The proposed OGWO algorithm provides a better outcome than the compared algorithms. OGWO achieves an overall efficacy of ≈45%, ≈30% and ≈20% greater than LEACH, ABC and GWO. In contrast to LEACH, OGWO renovates the bunching when a CH's death befalls and links the sensors to other CHs. The main

limitation of LEACH occurs as the CH's death occurs: the corresponding cluster becomes futile, and the collected information fails to transfer to the sink, i.e., BS. Moreover, LEACH picks the CH within the limit, which might lead to improper bunching and degrades the network's performance. In our work, OGWO also achieves the consistent dispersal of CHs. In addition, the optimal selection of CHs is considered as a significant process in improving the network's lifespan.

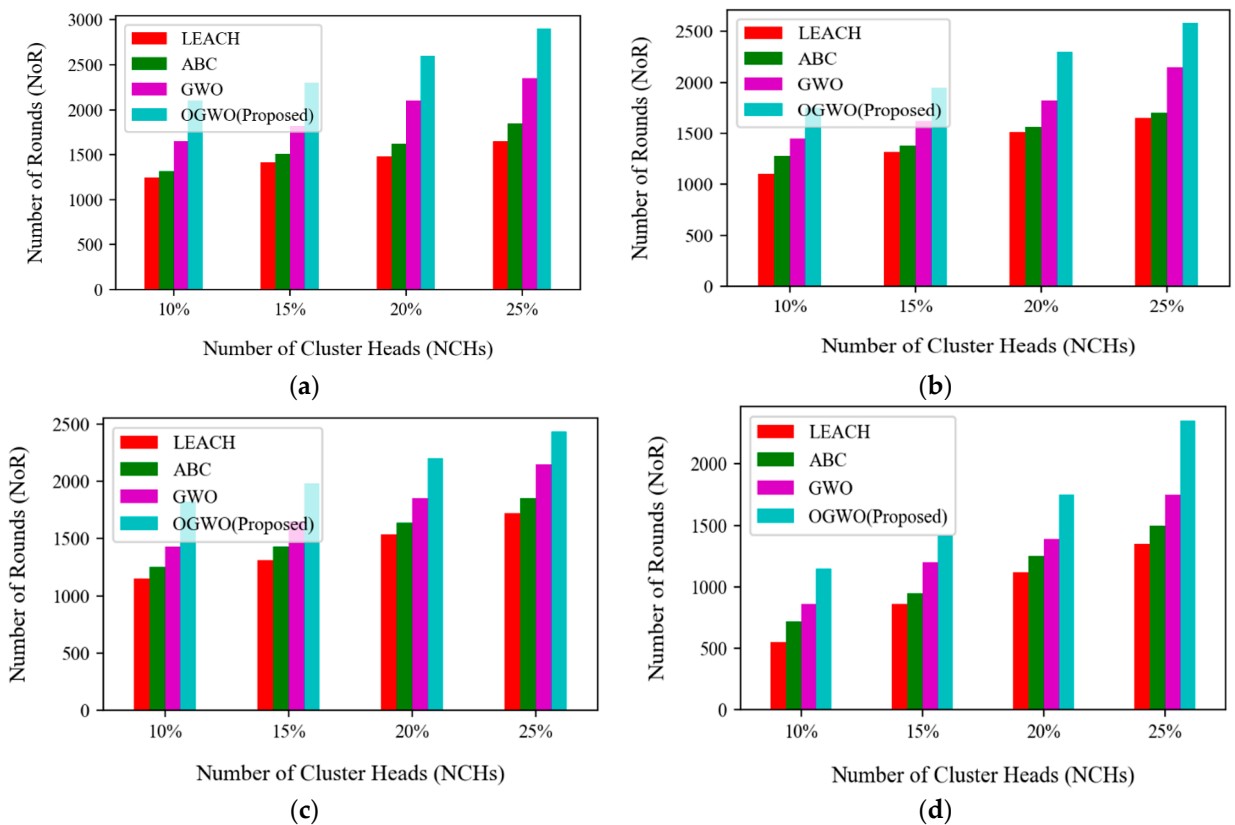

**Figure 4.** Comparison results of FND with respect to NoR by varying different CHs and NSNs (**a**) NSNs = 100, (**b**) NSNs = 200, (**c**) NSNs = 300, (**d**) NSNs = 400.

Our algorithm with a novel objective function performs well in determining the optimal CHs. Hence, the proposed OGWO algorithm achieves a better outcome than the LEACH technique. Furthermore, the existing metaheuristic algorithm of ABC and GWO fails to provide a better outcome because of an imbalance of intensification and diversification. Concisely, OGWO is an effective technique for identifying optimal CHs and improving the lifespan of the overall network.

*(iii) Test case 3 (NoPR):* In this test case 3, we used the deployment area as 200 × 200 m², BS location as ((0,0), (50,50), (100,100), and (150,150), initial energy as 2.0 J; the NSNs are considered as 100 to 400 nodes, and the NCHs are fixed as 10%. We used this test case to analyze the performance of the proposed algorithm in terms of the number of packets delivered at the BS. We graphically illustrated the observed results of the proposed algorithm and other techniques in Figure 6. Furthermore, we used a changing quantity of sensors from 100 to 400 concerning BS variations, respectively. From Figure 6, the results are measured based on the various number of sensor nodes: Figure 6a represents the output for NSNs = 100, Figure 6b for NSNs = 200, Figure 6c for NSNs = 300 and Figure 6d NSNs = 400. Similarly, the x-axis and y-axis represent the BS location and Number of Packets Received (NoPR).

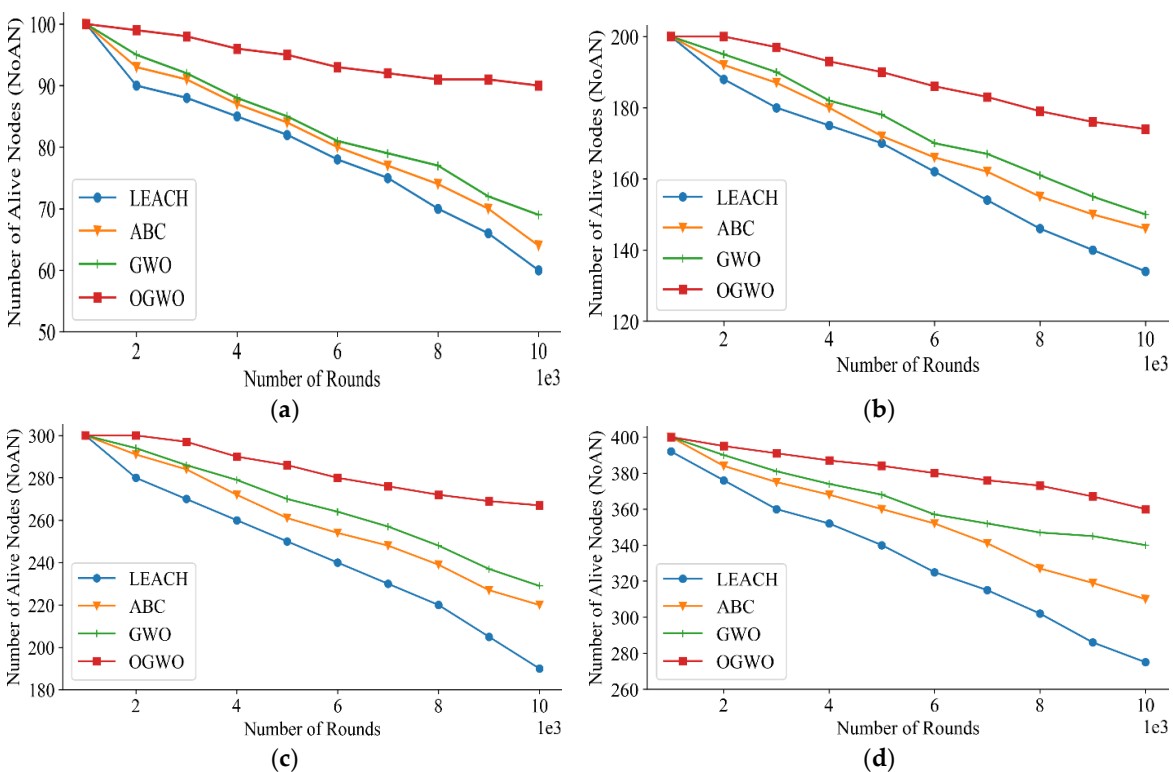

**Figure 5.** Comparison results of NoAN with respect to number of NCHs = 25%: (**a**) NoAN at NSN = 100 (**b**) NoAN at NSN = 200, (**c**) NoAN at NSN = 300, (**d**) NoAN at NSN = 400.

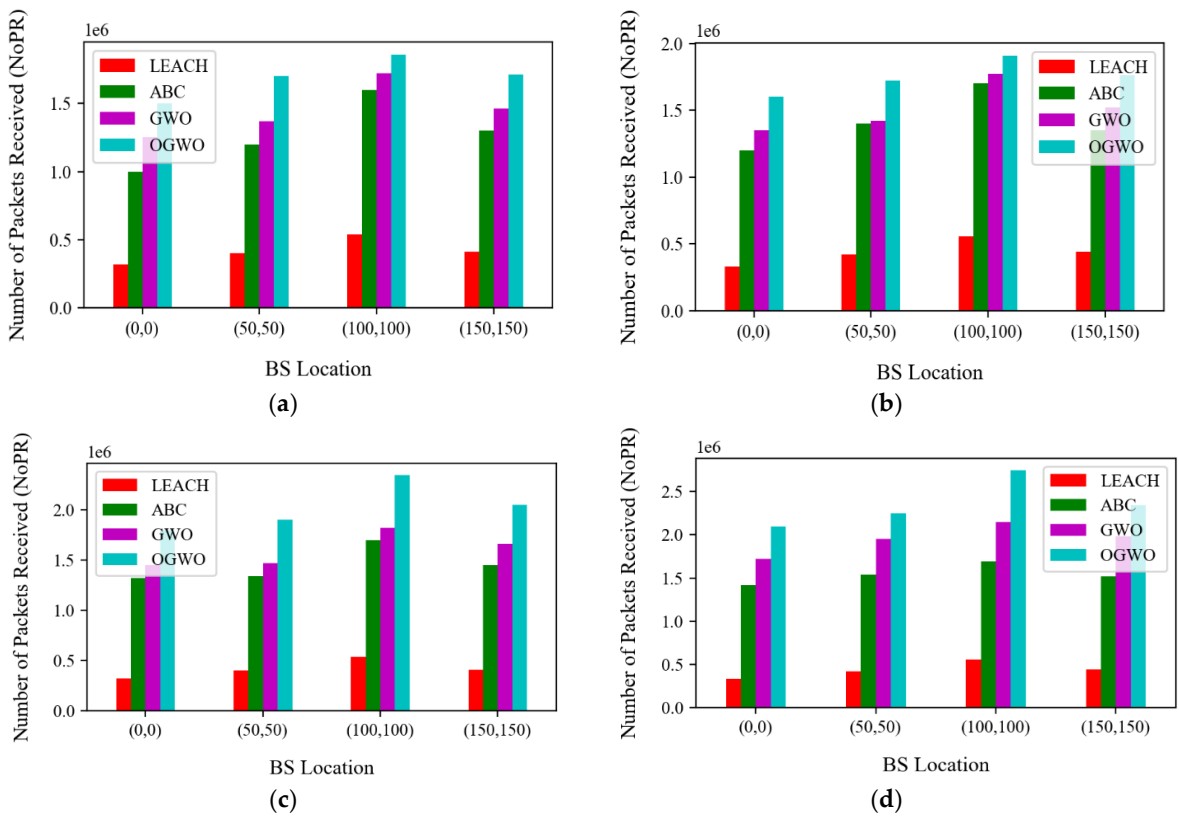

**Figure 6.** Comparison results of NoPR by BS concerning NCHs = 10%: (**a**) NoPR at NSN = 100, (**b**) NoPR at NSN = 200, (**c**) NoPR at NSN = 300, (**d**) NoPR at NSN = 400.

The proposed OGWO algorithm achieves a better outcome than the LEACH, ABC and GWO techniques. The achievement of OGWO is due to the efficient selection of CHs by processing unified diversity among the search agents compared with other algorithms. In addition, we observed that the network's lifespan is improved and consumes minimal energy to transmit the data packets to BS by electing the CHs within a minimal distance. Thereby, it increases the NoPR at the BS, although the position of BS is changeable. Our work provides a higher NoPR when the BS is located at (100,100), whereas the BS in other locations provides less NoPR than the BS position in the center region of the deployment area.

*(iv) Test case 4:* Our work considers this test case for analyzing the performance of the OGWO algorithm with respect to the impact of BS and NCHs. The parameters used in this work include a deployment area of $200 \times 200$ m$^2$ with the varying number of sensor nodes from 100 to 400 and an initial energy of 2.0 J. Initially, we used the NCHs as 25%, the number of sensor nodes as 100 to 400, and the number of BS positions as (0,0), (50,50), (100,100), and (150,150). The performance impact of the base station location concerning the FND over NoR is illustrated in Figure 7. The x-axis and y-axis epitomize the number of sensor nodes and NoR. The BS locations with (100,100) increase the survival of sensors for a certain number of rounds for all the varying node sensors, as shown in Figure 7. Concisely, the BS locations (0,0), (50,50) and (150,150) decreases the performance of the network more than BS location (100,100). This lack of performance occurs due to the far location of BS from the selected CHs. Therefore, the transmission of data packets from CHs might travel a maximum length to reach the BS at a distant location, leading to a minimal lifespan of the network. The BS at the center of the deployment area maximizes the lifespan and reduces the distance of data travel between the CHs and BS. In addition, the BS location at (100,100) improves the lifespan by ≈50% more than the other BS locations.

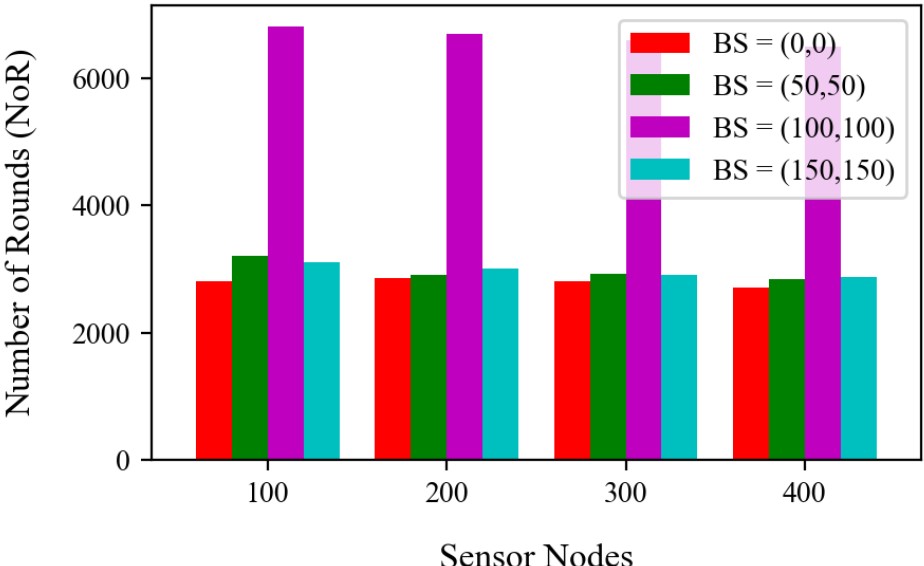

**Figure 7.** Impact of BS on the network lifetime.

Later, we set up a scenario with CHs of 10% to 25%, NSNs as 100 to 400 and BS location at (100,100) to analyze the concert impact due to the varying size of CHs as given in Figure 8. The x-coordinate and y-coordinate represent the number of sensors and rounds after the first node death. From Figure 8, we noticed that the number of iterations increases when the CHs have opted as 15% compared with CHs as 25%. Although the CHs with 25% provide better outcomes in the 100 and 200 sensor nodes environments, it degrades slowly when the number of the sensor increases to 300 and 400 nodes. Generally, CHs consume high energy compared to standard sensors because the CHs collect the data from the non-CHs, aggregate the information, and transmit the information to the BS. Therefore, the optimal CHs for the appropriate number of sensors is challenging. However, the low selection of

CHs will lead to high energy utilization and reduce the network's lifespan. Based on the analysis, we noticed that CHs with 15% provide adequate results compared with the CHs with 10%, 20% and 25%. In addition, the CHs with 15% increase the network's lifetime by holding the maximum number of rounds after the FND.

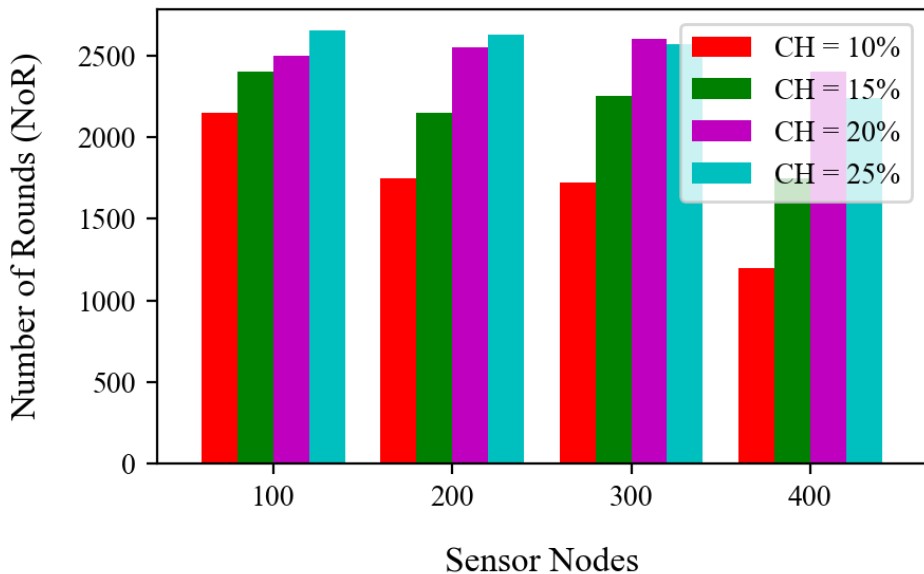

**Figure 8.** Impact of NCHs on the network lifetime.

The time complexity of our proposed algorithm is computed based on the number of function evaluations processed in determining the optimal CHs. For computing the node degree, it takes a time complexity of $O(\Delta)$, where $\Delta$ specifies the node degree. CH selection and cluster formation takes a complexity of $O(1)$. In addition, the data transfer from cluster member to CH required a complexity of $O(\Delta)$, whereas the data transfer from CH to BS is $O(\frac{d}{R})$, where R specifies the transmission range of each sensor and $d$ specifies the CHs distance from BS. The overall time complexity of our proposed work takes $O(\Delta * d/R)$.

## 6. Conclusions

This research work has aimed to introduce an optimal cluster head election framework by developing novel hybrid optimization techniques. We used different objective constraints for electing the optimal CHs such as residual energy and various distance metrics, node degree and node centrality. The formulated non-linear objective function has achieved the network's lifespan improvement. A novel hybrid technique, oppositional gray wolf optimization (OGWO), has been proposed by incorporating generic GWO and opposition-based learning techniques. This approach enriches the limitations of the existing algorithm by balancing the intensification and diversification of search agents in electing the optimal CHs. We implemented our proposed work in MATLAB 2018a with an adequate simulation environment. The experimental results suggest that our proposed algorithm provides a better outcome in improved network lifespan. The proposed OGWO algorithm attains the overall network lifespan of 45%, 30% and 20% over LEACH, ABC and GWO techniques. In addition, we also analyzed the impact of varied BS locations and CHs percentage concerning the different number of sensor nodes. We noticed that the improvement of networks lifespan depends on the position of the BS and the portion of CHs in the network. In future, we plan to use various GWO variants to strengthen the CH selection framework in a heterogeneous WSN. Furthermore, we compare and analyze the performance of different GWO variants to such issues; we also try different approaches to reduce the computation time and prolong the network lifespan.

**Author Contributions:** Conceptualization, R.R.; methodology, D.K. and A.B.; validation, M.R. and S.S.A.; formal analysis, A.D. and A.A.; writing—original draft preparation, D.K.; writing—review and editing, A.B. and M.R.; supervision, A.D. and R.R.; funding acquisition, A.A. All authors have read and agreed to the published version of the manuscript.

**Funding:** This study was funded by the Deanship of Scientific Research, Taif University Researchers Sup-porting Project number (TURSP-2020/148), Taif University, Taif, Saudi Arabia.

**Institutional Review Board Statement:** Not applicable.

**Informed Consent Statement:** Not applicable.

**Data Availability Statement:** Data in this research paper will be shared upon request made to the corresponding author.

**Acknowledgments:** The authors would like to thank the Deanship of Scientific Research, Taif University Researchers Supporting Project number (TURSP-2020/148), Taif University, Taif, Saudi Arabia for supporting this research work.

**Conflicts of Interest:** The authors declare no conflict of interest.

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
