# Peer review of "OGWO-CH: Hybrid Opposition-Based Learning with Gray Wolf Optimization Based Clustering Technique in Wireless Sensor Networks"

_electronics, doi:10.3390/electronics11162593_

Round 1
Reviewer 1 Report
This article introduces an ideal cluster head selection framework using hybrid Opposition-based learning with Grey Wolf Optimization (OGWO) algorithm. From my view, this paper is well organized and the proposed method is valuable for this research filed. After reviewed this paper, there are some questions and suggestions as follows.
- Some figures need to be enhanced in terms of quality and resolution.
- Abstract section needs to be re-drafted to be self-contained means it has to clearly show the hypothesis, methodology, techniques and tools used, and the results obtained.
- It is necessary to talk about the role of the parameters of the proposed algorithm in a separate section. For example: -Which parameters are responsible for controlling exploration and which parameters are responsible for controlling exploitation?
- You must review all significant similar works that have been done. Also, review some of the good recent works that have been done in this area and are more similar to your paper. For example: a) GGSA: A grouping gravitational search algorithm for data clustering, b) Using gravitational search algorithm for finding near-optimal base station location in two-tiered WSNs, c) An energy aware grouping memetic algorithm to schedule the sensing activity in WSNs-based IoT for smart cities, d) Gravitational search algorithm to solve the K-of-N lifetime problem in two-tiered WSNs.
- The comparison section is relatively weak. The proposed method should be compared with at least 3 other novel metaheuristic methods.
- The experimental results indicate that they perform well, but providing a stronger theoretical analysis and justification for the algorithm would be more convincing. To clearly state the objective of the research in terms of problems to address and expected results and show how the proposed technique will advance the state of the art by overcoming the limitations of the existing work. Also, the results obtained must be interpreted.
- It is necessary to experimentally analyze the proposed algorithm in terms of time consumed and compare with other algorithms.
- What are the advantages and disadvantages of this study compared to the existing studies in this area?
- There are some grammatical mistakes and typo errors.
Some final cosmetic comments:
* The results of your comparative study should be discussed in-depth and with more insightful comments on the behaviour of your algorithm on various case studies. Discussing results should not mean reading out the tables and figures once again.
* Avoid lumping references as in [x, y] and all other. Instead summarize the main contribution of each referenced paper in a separate sentence. For scientific and research papers, it is not necessary to give several references that say exactly the same. Anyway, that would be strange, since then what is innovative scientific contribution of referenced papers? For each thesis state only one reference.
* Avoid using first person.
* Avoid using abbreviations and acronyms in title, abstract, headings and highlights.
* Please avoid having heading after heading with nothing in between, either merge your headings or provide a small paragraph in between.
* The first time you use an acronym in the text, please write the full name and the acronym in parenthesis. Do not use acronyms in the title, abstract, chapter headings and highlights.
* The results should be further elaborated to show how they could be used for the real applications.
* Are all the images used in this work copyrights free? If not, have the authors obtained proper copyrights permission to re-use them? Please kindly clarify, and this is just to ensure all the figures are fine to be published in this work.
* Also, the list of references should be carefully checked to ensure consistency with between all references and their compliances with the journal policy on referencing.
Author Response
Dear Reviewer,
Thank you for spending time on our manuscript and providing us an opportunity for submitting the revised draft of our manuscript. We would like to acknowledge you for your comments and suggestions that helped us improve the manuscript. We have considered the comments with care and addressed them with the best of our efforts and knowledge. Each comment is point by point answered in attachment for your consideration.

Reviewer 2 Report
The manuscript entitled “OGWO-CH: Hybrid Opposition-based learning with Grey Wolf Optimization Based Clustering Technique in Wireless Sensor Networks” has been investigated in detail. The topic addressed in the manuscript is potentially interesting and the manuscript contains some practical meanings, however, there are some issues which should be addressed by the authors:
1) In the first place, I would encourage the authors to extend the abstract more with the key results. As it is, the abstract is a little thin and does not quite convey the interesting results that follow in the main paper. The "Abstract" section can be made much more impressive by highlighting your contributions. The contribution of the study should be explained simply and clearly.
2) The readability and presentation of the study should be further improved. The paper suffers from language problems.
3) The “Introduction” section needs a major revision in terms of providing more accurate and informative literature review and the pros and cons of the available approaches and how the proposed method is different comparatively. Also, the motivation and contribution should be stated more clearly.
4) The importance of the design carried out in this manuscript can be explained better than other important studies published in this field. I recommend the authors to review other recently developed works.
5) What makes the proposed method suitable for this unique task? What new development to the proposed method have the authors added (compared to the existing approaches)? These points should be clarified.
6) “Result analysis and Discussion” section should be edited in a more highlighting, argumentative way. The author should analysis the reason why the tested results is achieved.
7) The authors should clearly emphasize the contribution of the study. Please note that the up-to-date of references will contribute to the up-to-date of your manuscript. The study named "Metaheuristic optimization-based path planning and tracking of quadcopter for payload hold-release mission; Digital currency forecasting with chaotic meta-heuristic bio-inspired signal processing techniques " - can be used to explain the proposed optimization method in the study or to indicate the contribution in the “Introduction” section.
8) The complexity of the proposed model and the model parameter uncertainty are not enough mentioned.
9) It will be helpful to the readers if some discussions about insight of the main results are added as Remarks.
This study may be proposed for publication if it is addressed in the specified problems.
Author Response

(The authors gave the same response as above.)

Round 2
Reviewer 1 Report
Good revisions have been made in the paper and the revised version has the necessary qualities for acceptance compared to the previous version. In my opinion, the article is acceptable in its current form.
Reviewer 2 Report
All my comments have been thoroughly addressed. It is acceptable in the present form.